# Comparison of Anthropogenic Aerosol Climate Effects among Three Climate Models with Reduced Complexity

**Xiangjun Shi** [1,2,*], **Wentao Zhang** [1] **and Jiaojiao Liu** [1]

1   School of Atmospheric Sciences, Nanjing University of Information Science and Technology, Nanjing 210044, China
2   International Pacific Research Center, University of Hawaii at Manoa, Honolulu, HI 96822, USA
*   Correspondence: shixj@nuist.edu.cn

**Abstract:** The same prescribed anthropogenic aerosol forcing was implemented into three climate models. The atmosphere components of these participating climate models were the GAMIL, ECHAM, and CAM models. Ensemble simulations were carried out to obtain a reliable estimate of anthropogenic aerosol effective radiative forcing (ERF). The ensemble mean ERFs from these three participating models with this aerosol forcing were −0.27, −0.63, and −0.54 W·m$^{-2}$. The model diversity in ERF is clearly reduced as compared with those based on the models' own default approaches (−1.98, −0.21, and −2.22 W·m$^{-2}$). This is consistent with the design of this aerosol forcing. The modeled ERF can be decomposed into two basic components, i.e., the instantaneous radiative forcing (RF) from aerosol–radiation interactions (RFari) and the aerosol-induced changes in cloud forcing (△Fcloud$^{*}$). For the three participating models, the model diversity in RFari (−0.21, −0.33, and −0.29 W·m$^{-2}$) could be constrained by reducing the differences in natural aerosol radiative forcings. However, it was difficult to figure out the reason for the model diversity in △Fcloud$^{*}$ (−0.05, −0.28, and −0.24 W·m$^{-2}$), which was the dominant source of the model diversity in ERF. The variability of modeled ERF was also studied. Ensemble simulations showed that the modeled RFs were very stable. The rapid adjustments (ERF − RF) had an important role to play in the quantification of the perturbation of ERF. Fortunately, the contribution from the rapid adjustments to the mean ERF was very small. This study also showed that we should pay attention to the difference between the aerosol climate effects we want and the aerosol climate effects we calculate.

**Keywords:** aerosol climate effects; reduced complexity; model diversity; calculation methods

## 1. Introduction

Anthropogenic aerosols are thought to be responsible for the second largest source of anthropogenic radiative perturbations [1–5]. More importantly, the magnitude of anthropogenic aerosols' contribution is the dominant source of uncertainty in estimating the planetary energy imbalance caused by human activities [6–9].

Climate models are an important tool for studying how anthropogenic aerosols affect the climate system. Thus, significant progress was made in the development of aerosol-process treatments and aerosol–cloud interaction treatments in climate models over the last two decades [10–13]. Using these climate models, aerosol optical and cloud-interaction properties can be calculated and used to estimate aerosol climate effects. This approach proved to be challenging because important aerosol processes and aerosol–cloud interactions remain poorly understood [14–16], and even well-understood processes are difficult to simulate consistently in large-scale climate models [17–19]. As a result, individual climate

models differ widely in estimating aerosol effects [4,14,20,21]. To reduce this kind of uncertainty in climate models, a given distribution of anthropogenic aerosol direct radiative forcing and an associated Twomey effect are recommended by the Coupled Model Intercomparison Project Phase 6 (CMIP6). To facilitate this approach, Stevens et al. [22] introduced a simple parameterization to the second version of the Max Planck Institute Aerosol Climatology (MACv2-SP), which prescribes anthropogenic aerosol optical properties and the effect of anthropogenic aerosols on the cloud droplet number concentration. In this study, the MACv2-SP was implemented into three climate models. It was supposed that the differences in estimates of anthropogenic aerosol climate effects among these three participating models should be reduced as compared with the models' own default approaches [2,7]. One goal of this study is to test this hypothesis. Furthermore, the still existing differences are also discussed to better understand the sources of the model diversity in estimating anthropogenic aerosol effects.

Model simulations from CMIP5 and previous CMIP phases showed that the response to total external forcings (e.g., solar variability, greenhouse gases, and aerosols) differ widely among different models. In order to identify reasons for the differences, it is necessary to quantify radiative forcing from various external forcing factors in each model [23,24]. CMIP6 encourages all modeling groups to participate in the Radiative Forcing Model Intercomparison Project (RFMIP) [25]. Simulation results from RFMIP can be used to diagnose anthropogenic aerosol climate effects [25]. However, the aerosol experimental design in RFMIP does not distinguish aerosol–radiation interactions (abbreviated "ari") from aerosol–cloud interactions (abbreviated "aci"), where the latter contains the Twomey effect and its subsequent rapid adjustments to radiative forcing. It is important to note that the uncertainty in the anthropogenic aerosol Twomey effect from MACv2-SP is higher than the aerosol optical properties [22], and that the representation of "aci" in climate models is much more complex than "ari" [16,26–29]. Furthermore, distinguishing the contributions to aerosol forcing from "ari" and from "aci" was historically essential for understanding the associated mechanisms [27,30–32]. Therefore, it is necessary to estimate the anthropogenic aerosol effects from "ari" and "aci" separately. This study is designed to fill the gap for understanding model differences in estimating anthropogenic aerosol effects.

Recent studies showed that anthropogenic aerosol instantaneous radiative forcing (RF) can be well constrained after simulating just one year, whereas it is difficult to get a very stable estimate of anthropogenic aerosol effective radiative forcing (ERF) [29,33]. Thus, it is necessary to present the RF and the uncertainty in the magnitude of ERF. Note that the RF caused by "ari" can be diagnosed from the difference between two radiation calls with and without aerosol optical properties at each radiation time step. However, it is often hard to estimate the RF caused by the Twomey effect without significant model modifications. As a result, the RF shown in previous studies usually only presented the RF from "ari" [32]. In this study, one participating climate model is modified to diagnose the RF caused by the Twomey effect. This helps us to identify the uncertainty sources in calculating the ERFs from the Twomey effect that stay hidden in the model complex.

In this study, anthropogenic aerosol effects are simulated by three participating climate models with reduced complexity (i.e., the MACv2-SP). Section 2 introduces the models, the experiment strategy, and the methods for calculating anthropogenic aerosol effects. The MACv2-SP parameterization is also briefly introduced in this section. Section 3 presents the estimate of anthropogenic aerosol climate effects. The conclusions and discussions are presented in Section 4.

## 2. Models, Methods, and Experiments

### 2.1. MACv2-SP

The anthropogenic aerosol forcing officially recommended by CMIP6 is a hypothetical dataset derived from the second version of the Max Planck Institute Aerosol Climatology (MACv2-SP). The MACv2-SP parameterization calculates the spatio-temporal distribution and wavelength dependence of anthropogenic aerosol optical properties, which includes wavelength-dependent aerosol optical depth, single-scattering albedo, and asymmetry factor. Note that the longwave effect

of anthropogenic aerosols in radiation transfer is neglected. The anthropogenic aerosol forcing from aerosol–cloud interactions ("aci") only considers the Twomey effect. The Twomey effect is represented by the normalized change in cloud droplet number (dN), which ensures that the proportional change in cloud droplet number concentration (N) due to anthropogenic aerosols is insensitive to background N. The background N provided by the host model indicates the N induced by natural aerosols. Note that the multiplicative factor (i.e., dN) only works on the N used for calculating cloud optical properties. Additional details about MACv2-SP can be found in Stevens et al. [22].

## 2.2. Participating Models

In this study, the MACv2-SP parameterization was implemented into three climate models, which are run here with an atmosphere-only mode. The atmosphere components of these participating climate models are the latest version of the Grid-Point Atmospheric Model of IAP LASG (hereafter, GAMIL), the modified Max Planck Institute's ECHAM model version 6.3 (hereafter, ECHAM), and version 5.3 of the Community Atmosphere Model (hereafter, CAM).

The GAMIL model is the atmospheric component of the Flexible Global Ocean–Atmosphere–Land System Model developed by the Institute of Atmospheric Physics, Chinese Academy of Sciences (FGOALS) [34–36]. The GAMIL model with MACv2-SP parameterization is used in CMIP6. The Delta–Eddington Approximation developed by Briegleb [37] is used for the solar radiation flux transfer calculations. The solar spectrum is divided into 19 discrete spectral and pseudo-spectral intervals, following Collins [38]. The aerosol direct radiative effect is represented by globally uniform natural aerosol optical properties. Now, anthropogenic aerosol optical properties provided by MACv2-SP are considered. Several years ago, the GAMIL model was updated to include a two-moment cloud microphysics scheme and a physically based aerosol activation parameterization [39,40]. The physically based aerosol activation parameterization, which calculates the cloud condensation nuclei (CCN), is derived from a dataset of prescribed aerosols. The N is set to the number of active CCN if the N decreases below that number. In this study, background N is calculated based on pre-industrial (PI, year 1850) prescribed aerosol data. To consider the anthropogenic aerosol Twomey effect based on the CMIP6 protocol, the background N used for calculating cloud optical properties is multiplied by the multiplicative factor (i.e., dN), which is calculated from MACv2-SP.

Here, the modified ECHAM model is used as the atmospheric component of the Nanjing University of Information Science and Technology (NUIST) Earth System Model version 3 (NESM), which was registered for CMIP6 [41]. In this model, radiative transfer is solved using optimized two-stream Rapid Radiative Transfer Model for General circulation models (RRTMG) developed by Atmospheric and Environmental Research of Lexington, Massachusetts [42]. The input prescribed aerosol optical properties used for radiative transfer calculations were separated into coarse and fine modes. Coarse-mode aerosol climatology data are assumed to be of natural origin, composed of dust and sea salt. Fine-mode aerosol is a combination of natural and anthropogenic aerosols, such as sulfate and organic matter including black carbon. The aerosol optical properties of fine-mode aerosols affect 14 solar spectral bands, whereas the interaction of these small particles with light in the thermal spectral range is negligible. In this study, the default fine-mode aerosol optical property data from the year 1850 are taken as the natural contribution and are combined with anthropogenic aerosol properties calculated by MACv2-SP. Stratiform clouds in the model are represented by a one-moment cloud microphysical scheme developed by Lohmann and Roeckner [43]. The N is a prescribed value that depends on surface land-use type and vertical layer pressure. To determine the anthropogenic aerosol Twomey effect, the prescribed N used for the radiation scheme is multiplied by the dN from MACv2-SP.

The CAM model is the atmospheric component of the Community Earth System Model (CESM) developed by NCAR (National Center for Atmospheric Research). In this model, a two-moment stratiform cloud microphysics scheme [44,45] is used and coupled to a modal aerosol module [11,46] for considering "aci". The RRTMG radiation package is used to more accurately represent aerosol and cloud effects [47]. To estimate aerosol effects based on the CMIP6 protocol, PI (year 1850) aerosol

and precursor emissions are applied to derive the CAM model. The online calculated aerosol optical properties and cloud optical properties based on these PI aerosol data are assumed to be the natural aerosol contribution. The aerosol optical properties for 14 shortwave spectral bands derived from MACv2-SP are added to the natural aerosol optical properties. Similarly, with the GAMIL and ECHAM models, cloud optical properties are calculated from the amplified N to present the Twomey effect.

### 2.3. Description of Experiments

All experiments are carried out with the atmosphere-only model configurations using prescribed sea surface temperature and sea-ice concentrations. The GAMIL model is conducted at a horizontal resolution of $80 \times 180$ grids and 26 vertical levels. The ECHAM model employs a horizontal resolution of $96 \times 192$ grids and 47 vertical levels. CAM model simulations run at a horizontal resolution of $96 \times 144$ grids and 30 vertical levels.

There were five sensitivity experiments conducted in this study, named CTL, ARI, ACI, ALL, and OLD (Table 1). The CTL, ARI, ACI, and ALL experiments were driven by the models' own natural aerosol data (PI, year 1850) and anthropogenic aerosol forcings from MACv2-SP. The CTL experiment ran with anthropogenic aerosol forcings for the year 1850 (i.e., no anthropogenic aerosol). Compared to the CTL experiment, the present-day (PD, year 2000) anthropogenic aerosol optical properties were added in the ARI experiment. The ACI experiment included the PD anthropogenic aerosol Twomey effect as compared to the CTL experiment. As compared to the CTL experiment, the PD anthropogenic aerosol optical properties and the Twomey effect based on the CMIP6 protocol were considered in the ALL experiment. In the OLD experiment, models were driven by their own PD aerosol data and treated the anthropogenic aerosol forcings based on their own default mechanisms.

**Table 1.** List of sensitivity experiments conducted in this study.

| Names | Description |
| --- | --- |
| CTL | Simulation using model default pre-industrial times (PI, the year of 1850) aerosol forcings (i.e., anthropogenic aerosol forcings are excluded). |
| ARI | Same as CTL, but present-day (PD, the year of 2000) anthropogenic aerosol optical properties from the second version of the Max Planck Institute Aerosol Climatology (MACv2-SP) are considered. |
| ACI | Same as CTL, but PD anthropogenic aerosol Twomey effect from MACv2-SP is considered. |
| ALL | Same as CTL, but PD anthropogenic aerosol optical properties and Twomey effect from MACv2-SP are considered. |
| OLD | Same as CTL, but with model default PD aerosol forcings. |

The effective radiative forcing (ERF) is obviously influenced by model internal variability, and it is difficult to get a stable estimate [29,33]. Thus, we produced ensemble simulations in which each experiment had 10 simulations starting in different months. For the same participating model, the same set of 10 different start dates was used for all experiments. Taking CAM model experiments for example, the first ensemble member in each experiment (i.e., the CTL, ARI, ACI, ALL, and OLD experiments) started on 1 January, the second ensemble member in each experiment started on 1 February, and so on. All simulations were run for ~11–12 years, and results from the last 10 years (from January to December) were used in the analysis, while only the difference between two simulations with the same start date was analyzed. For instance, we calculated a 10-year average ERF estimate for each pair of the 10 ensemble members, i.e., 10 values per model (10 differences between two experiments, first–first, second–second, and so on). The standard deviation, which was used for variability analysis, was calculated from these 10 values. It is noteworthy that, in the same experiment, the differences in 10-year average radiative fluxes between two ensemble members (i.e., two simulations with the different initialization date) were notable due to model internal variability. It is better to exclude these

kinds of differences from estimating ERF. This is the reason why pairs of experiments with the same initialization date were used for differences.

*2.4. Methods Used to Estimate Aerosol Effects*

In this study, instantaneous radiative forcing (RF), effective radiative forcing (ERF), and their uncertainties were calculated to estimate anthropogenic aerosol effects. In order to comprehend the estimate deeply, it is necessary to explain the diagnosed variables and their calculation methods clearly. We used the capital letter "F" to indicate the all-sky shortwave net radiative fluxes at the top of the atmosphere (TOA). The F marked with the superscript "*" (i.e., $F^*$) was diagnosed from radiation call with aerosol scattering and absorption neglected. The F marked with the superscript "c" (i.e., $F^c$) is the clear-sky F, which was diagnosed from radiation call without cloud effect. The shortwave aerosol forcing and shortwave cloud forcing were named Faerosol and Fcloud, respectively. Faerosol = $F - F^*$ and Fcloud = $F - F^c$. For the convenience of readers, all model output variables analyzed in this study are listed in Table 2.

**Table 2.** List of model output variables analyzed in this study.

| Names | Description |
|---|---|
| F (W·m$^{-2}$) | The all-sky shortwave net radiative fluxes at the top of the atmosphere (TOA). |
| $F^c$ (W·m$^{-2}$) | The clear-sky F. |
| $F^*$ (W·m$^{-2}$) | Same as F, but calculated from radiation call without aerosol radiative effect. |
| $F^{c*}$ (W·m$^{-2}$) | The clear-sky $F^*$. |
| $F^\#$ (W·m$^{-2}$) | Same as F, but derived from cloud optical properties without aerosol Twomey effect. |
| Faerosol (W·m$^{-2}$) | The shortwave aerosol forcing, Faerosol = $F - F^*$. |
| Faerosol$^c$ (W·m$^{-2}$) | The clear-sky Faerosol, Faerosol$^c$ = $F^c - F^{c,*}$. |
| Fcloud (W·m$^{-2}$) | The shortwave cloud forcing, Fcloud = $F - F^c$. |
| Fcloud$^*$ (W·m$^{-2}$) | Fcloud without aerosol radiative effect, Fcloud$^*$ = $F^* - F^{c,*}$. |
| dFcloud (W·m$^{-2}$) | The impact of aerosol radiative effect on calculating Fcloud, dFcloud = Fcloud − Fcloud$^*$. |
| RFaci (W·m$^{-2}$) | The instantaneous aerosol Twomey effect, RFaci = $F - F^\#$. |
| AOD | The aerosol optical depth in the visible band. |
| AODa | The anthropogenic aerosol optical depth in the visible band calculated from MACv2-SP. |
| dN | The normalized change in drop number, calculated from MACv2-SP. |
| CLD (%) | The total cloud fraction. |
| LWP (g·m$^{-2}$) | The liquid water path. |
| CDNC ($10^{10}$ m$^{-2}$) | The background column-integrated grid-mean cloud droplet number concentration. |
| COD | The cloud optical depth in the visible band. |

[1] Note that only the GAMIL model diagnoses $F^\#$ and RFaci.

The ERF, which includes the impact of rapid adjustments, is recommended to estimate anthropogenic aerosol effects on the planetary energy balance [14,48]. These rapid adjustments are responses triggered by the forcing agent that are independent of surface temperature change. Here, the fixed SST (sea surface temperature) method was used to diagnose ERF. The change in longwave net radiative fluxes was small and neglected. One recommended way to calculate the ERF is using the top-of-atmosphere shortwave net radiative flux (F) difference between two simulations with and without anthropogenic aerosols [22,49,50]. The RF, which can be well constrained, is also widely used to analyze anthropogenic aerosol effects [4,25,33,48]. The difference between aerosol RF and ERF gives an estimate of atmospheric adjustments due to aerosol forcings [22,48].

Here, the ERF from aerosol–radiation interactions ("ari") and the Twomey effect (abbreviated ERFall) was calculated as $F^{ALL-CTL}$. In this paper, the superscript experiment name indicates that the model output variable comes from that experiment. For example, $F^{ALL}$ indicates the F derived from the ALL experiment, and $F^{ALL-CTL}$ indicates the difference in F between the ALL and CTL experiments. The ERFall in clear-sky conditions (abbreviated ERF$^c$all) was calculated as $F^{cALL-CTL}$, where $F^c$ is the clear-sky F. All these participating models can diagnose the instantaneous aerosol shortwave radiative forcing (Faerosol), which is the difference between the normal F and the F without

aerosol radiative effects ($F^*$), Faerosol $= F - F^*$. In this paper, the superscript "*" indicates that the radiative flux is calculated without considering aerosol radiative effects. Note that the Faerosol indicates the aerosol (anthropogenic and natural) radiative effects from "ari". The Faerosol$^{ALL-CTL}$ was used to estimate the anthropogenic aerosol RF from "ari" (abbreviated RFari). The RFari in clear-sky conditions (abbreviated RF$^c$ari) was approximated as Faerosol$^{cALL-CTL}$, where Faerosol$^c$ is the clear-sky Faerosol. Ghan [50] reported that anthropogenic aerosol "ari" has a significant impact on diagnosing the shortwave cloud radiative forcing (Fcloud), where Fcloud $= F - F^c$. In this paper, the superscript "c" indicates that the radiative flux is calculated without cloud optical properties. As recommended by his study, the anthropogenic aerosol effects on cloud radiative forcing are estimated as Fcloud$^{*ALL-CTL}$, where Fcloud$^* = F^* - F^{c*}$. Note that ERFall ($F^{ALL-CTL}$) = RFari(Faerosol$^{ALL-CTL}$) + Fcloud$^{*ALL-CTL}$ + F$^{c*ALL-CTL}$. In other words, the modelled ERFall can be decomposed into the contributions of RFari, Fcloud$^{*ALL-CTL}$, and F$^{c*ALL-CTL}$. The F$^{c*ALL-CTL}$ is usually very small and negligible as compared with RFari or Fcloud$^{*ALL-CTL}$ [50].

This paragraph introduces the method for estimating the ERF from "ari" (abbreviated ERFari). The ERFari was calculated as the difference in F between two simulations with and without anthropogenic "ari". There are two ways to calculate this difference, i.e., ARI − CTL and ALL − ACI. Both of them are shown in this study. The ERFari in clear-sky conditions (abbreviated ERF$^c$ari) was calculated as F$^{cARI-CTL}$ or F$^{cALL-ACI}$. In addition to the method introduced in the previous paragraph, the RFari can also be approximated by Faerosol$^{ARI-CTL}$ or Faerosol$^{ALL-ACI}$. Similarly, the RF$^c$ari can be approximated by Faerosol$^{cARI-CTL}$ or Faerosol$^{cALL-ACI}$. Rapid adjustments (i.e., ERFari − RFari) contribute to ERFari through clouds (i.e., semi-direct effect), the atmospheric profile, and surface energy budget [14]. Both Fcloud$^{*ARI-CTL}$ and Fcloud$^{*ALL-ACI}$ can be used to quantify the anthropogenic aerosol semi-direct effect. Note that ERFari ($F^{ARI-CTL}$) = RFari (Faerosol$^{ARI-CTL}$) + Fcloud$^{*ARI-CTL}$ + F$^{c*ARI-CTL}$ and ERFari ($F^{ALL-ACI}$) = RFari (Faerosol$^{ALL-ACI}$) + Fcloud$^{*ALL-ACI}$ + F$^{c*ALL-ACI}$. The difference in F$^{c*}$ between two experiments (i.e., F$^{c*ARI-CTL}$ or F$^{c*ALL-ACI}$) represents rapid adjustments induced by other ways. This difference is usually very small and negligible as compared with the difference in Fcloud$^*$ between two experiments (i.e., Fcloud$^{*ARI-CTL}$ or Fcloud$^{*ALL-ACI}$). Thus, the rapid adjustment from "ari" is often approximated by the semi-direct effect [14].

This paragraph introduces the method for estimating the ERF from "aci" (abbreviated ERFaci). The ERFaci was calculated as the difference in F between two simulations with and without anthropogenic "aci" (i.e., F$^{ACI-CTL}$ or F$^{ALL-ARI}$). Note that the aerosol indirect effects on warm clouds are often estimated by their impact on shortwave cloud radiative forcings, which is the difference in Fcloud or Fcloud$^*$ between two simulations [10,20,44,50–52]. The Twomey effect, also known as the cloud albedo effect, is defined as the anthropogenic aerosol acting to increase cloud droplet concentration and thereby the optical thickness with the liquid water content fixed [53]. In terms of this definition, the Twomey effect is the RF from aerosol–cloud interactions (abbreviated RFaci). The RFaci is a theoretical construct that is not easy to separate from subsequent rapid adjustments and is, therefore, rarely quantified [14]. In this study, the GAMIL model was modified to diagnose another dataset of cloud optical properties without the Twomey effect, and then another F was calculated with this dataset (F$^{\#}$). The RFaci was estimated by the difference between F and F$^{\#}$, RFaci $= F - F^{\#}$. The RFaci is a model output variable from one simulation rather than the difference between two simulations. It is important to note that the difference in Fcloud or Fcloud$^*$ between two simulations, which is often used to quantify the aerosol Twomey effect, actually accounts for the instantaneous Twomey effect (i.e., RFaci), as well as any secondary changes in cloud optical properties from rapid adjustments. As compared to the difference in Fcloud or Fcloud$^*$ between two simulations, ERFaci (i.e., the difference in F between two simulations) also includes subsequent changes in other optical factors from rapid adjustments.

In this study, the methods for calculating the RFs (i.e., RFall, RFari, and RFaci) and ERFs (i.e., ERFall, ERFari, and ERFaci) might not be unique. For example, the RFari can be approximated by Faerosol$^{ALL-CTL}$, Faerosol$^{ARI-CTL}$, or Faerosol$^{ALL-ACI}$. For the convenience of readers, we showed all

calculation methods for estimating ERF and its two basic components, i.e., the RF from aerosol–radiation interactions (RFari) and the aerosol-induced changes in cloud forcing (Fcloud* changes) in Section 4. Finally, it is noteworthy that all modeled aerosol climate effects (i.e., RFs, ERFs, and Fcloud* changes) were statistically tested. The non-significant results at the 10% level of the Student's *t*-test were deemed not robust, following Fiedler et al. [33].

## 3. Results

### 3.1. Anthropogenic Aerosol Forcings Used in Participating Models

First of all, we compared the anthropogenic aerosol forcings used in this study with the work of Stevens et al. [22]. The anthropogenic aerosol optical depth (AODa, 550 nm) for September 2005 and the droplet number multiplicative factor (dN) for 2005 shown by Stevens et al. [22] could be reproduced by the three models (not shown). In this study, the PD period was set to the year 2000 rather than 2005 because the year of the PD aerosol emission dataset for the CAM model was 2000. The AODa and dN for the year 2000 were similar to those for 2005 (not shown).

Figure 1 shows the PD AODa and dN from MACv2-SP. The global annual mean AODa in the visible bands used with the GAMIL, ECHAM, and CAM models were 0.032, 0.025, and 0.027, respectively. Note that the radiation package of GAMIL (Delta–Eddington Approximation) is different to that of ECHAM and CAM (RRTMG). As a result, the visible band of GAMIL (350.0–640.0 nm) was different to that of ECHAM and CAM (441.5–625.0 nm). This explains why AODa from GAMIL was slightly higher than that from ECHAM and CAM. Note that aerosol optical properties used in the GAMIL and CAM models were calculated at each model time step and were based on model grid vertical information, whereas the ECHAM model used a monthly input aerosol dataset. The vertical resolution of this input aerosol dataset was 500 m. This coarse near-surface vertical resolution caused the AODa used in the ECHAM model to be slightly smaller over the Tibetan Plateau than that in the other models. The dN is independent of vertical resolution or waveband range. Thus, the values of dN used for the GAMIL, ECHAM, and CAM models were extremely similar and had the same global annual mean value (1.075). Overall, the anthropogenic aerosol forcings used in the GAMIL, ECHAM, and CAM models were almost identical, especially for dN.

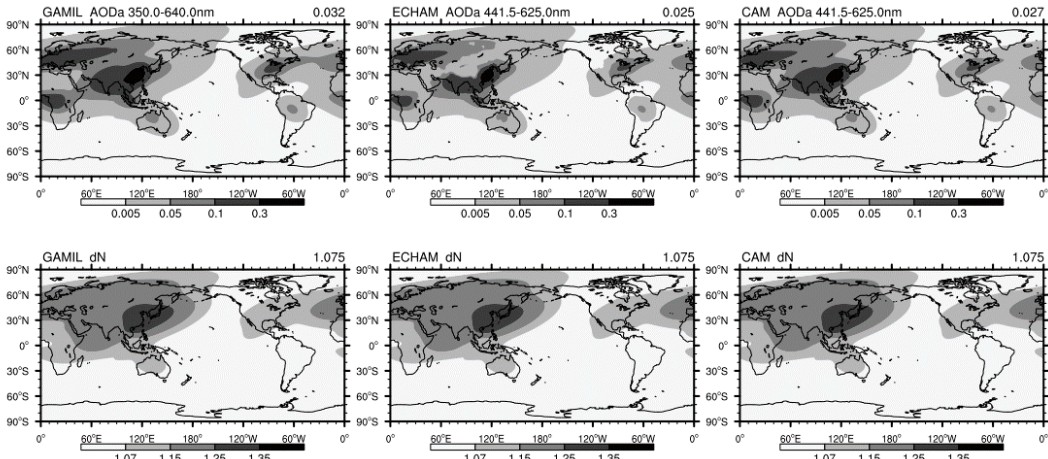

**Figure 1.** Anthropogenic aerosol (the year of 2000) optical depth (AODa, unitless, upper) and normalized change in drop number (dN unitless, lower) with the GAMIL (**left**), ECHAM (**middle**), and CAM (**right**) models. Global mean values are shown in the upper right corners.

### 3.2. The Climate Effects of Anthropogenic Aerosols

Figure 2 shows anthropogenic aerosol effective radiative forcing (ERFall = $F^{ALL-CTL}$) and its two main components, i.e., RF from direct radiative forcing (RFari = $Faerosol^{ALL-CTL}$) and effects on cloud

radiative forcing (Fcloud$^{*ALL-CTL}$). The global mean ERFalls from the GAMIL, ECHAM, and CAM models were −0.27, −0.63, and −0.54 W·m$^{-2}$, respectively. The ERFall from the GAMIL model was clearly weaker (less negative) than that from the ECHAM and CAM models. The global mean RFari from GAMIL was −0.21 W·m$^{-2}$, which was also weaker than that from the ECHAM (−0.33 W·m$^{-2}$) and CAM (−0.29 W·m$^{-2}$) models. However, the regional distribution pattern of RFari was generally similar for all models. As expected from the regional maximum in AODa (Figure 1), east and south Asia were the largest contributors to globally averaged RFari. The global means of Fcloud$^{*ALL-CTL}$ from the GAMIL, ECHAM, and CAM models were −0.05, −0.28, and −0.24 W·m$^{-2}$, respectively. As compared to RFari, the difference in Fcloud$^{*ALL-CTL}$ among these models contributed most to the model diversity in ERFall. For all models, there were few regions of Fcloud$^{*ALL-CTL}$ that could pass the 10% significance level test. In addition to the instantaneous Twomey effect (i.e., RFaci), Fcloud$^{*ALL-CTL}$ also includes the rapid adjustments from "ari" and the Twomey effect. The complex rapid adjustments might be the main source of uncertainty. In contrast, the diagnosed Faerosol can be well constrained because it is not closely related to rapid adjustments [25,33]. The RFari was statistically significant over high anthropogenic aerosol burden areas (AODa > 0.1). Because of the uncertainty in Fcloud$^{*ALL-CTL}$, ERFall was generally not statistically significant, expect for in some regions of east and south Asia.

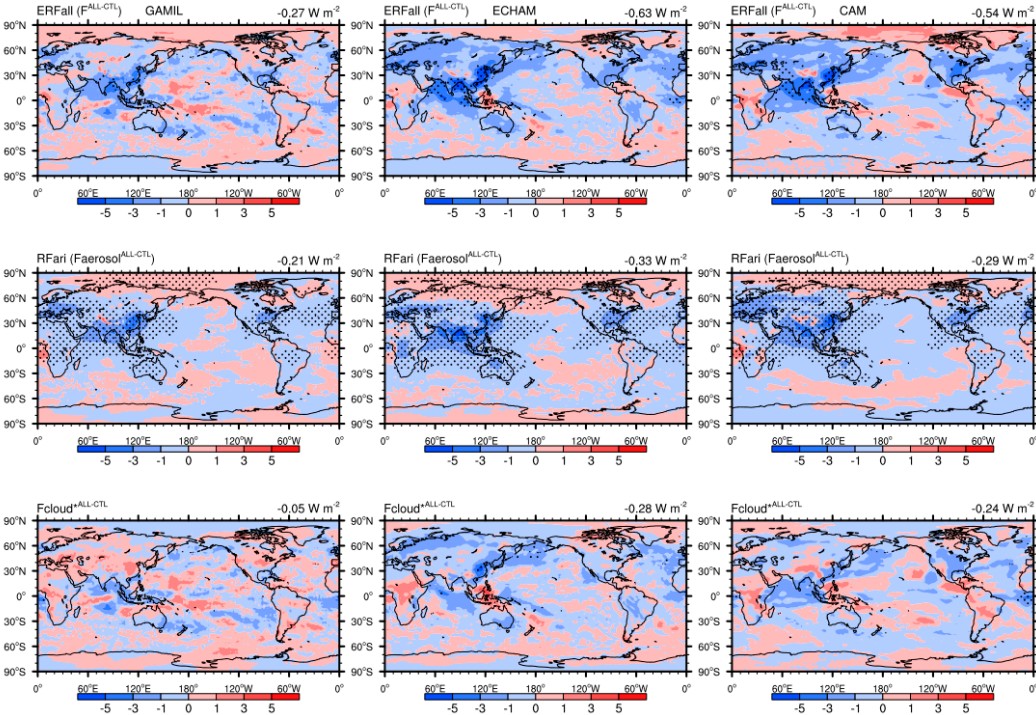

**Figure 2.** Multi-member ensemble means of anthropogenic aerosol effective radiative forcing (ERFall = F$^{ALL-CTL}$, upper panel), instantaneous radiative forcing from aerosol–radiation interactions (RFari = Faerosol$^{ALL-CTL}$, middle panel), and effects on shortwave cloud forcing (Fcloud$^{*ALL-CTL}$, lower panel) with the GAMIL (left), ECHAM (middle), and CAM (right) models. The number on the top right of the figure denotes the ensemble average of global annual mean values. Differences significant at the 10% level of the Student's *t*-test are depicted by dots.

Figure 3 shows the clear-sky effective radiative forcing (ERF$^c$all = F$^{cALL-CTL}$) and clear-sky instantaneous radiative forcing from "ari" (RF$^c$ari = Faerosol$^{cALL-CTL}$). For all models, the ERF$^c$all was almost the same as the RF$^c$ari. The global annual mean difference between ERF$^c$all and RF$^c$ari (i.e., F$^{c*ALL-CTL}$) was small and negligible (≤0.02 W·m$^{-2}$, Table 3). In other words, the RF$^c$ari clearly dominated the ERF$^c$all magnitude. The spatial patterns of negative RF$^c$ari were all roughly consistent with AODa (Figure 1), as RF$^c$ari depends primarily on "ari". The global means of RF$^c$ari from the GAMIL, ECHAM, and CAM models were estimated at −0.45, −0.73, and −0.74 W·m$^{-2}$, respectively.

The RF$^c$ari from GAMIL was substantially weaker than that from the ECHAM and CAM models. This result is consistent with RFari (Figure 2). Note that the clear-sky RF$^c$ari from CAM was a little stronger than that of ECHAM, whereas the all-sky RFari from CAM was a little weaker than that of ECHAM. This suggests that the cloud masking effect (RFari − RF$^c$ari) in the CAM model was stronger due to its higher cloud optical property (COD, Table 3). The statistically significant regions of negative RF$^c$ari were substantially larger than seen for RFari (Figure 2). After excluding the cloud masking effect, the RF$^c$ari was considerably less variable than the RFari.

**Table 3.** Ensemble averages of global annual mean results from the CTL, ALL, and OLD experiments. The details on these variables are listed in Table 2. Standard deviations (in brackets) are calculated from the different ensemble members.

| | GAMIL | | | ECHAM | | | CAM | | |
|---|---|---|---|---|---|---|---|---|---|
| | CTL | ALL −CTL | OLD −CTL | CTL | ALL −CTL | OLD −CTL | CTL | ALL −CTL | OLD −CTL |
| F | 237.98 | −0.27 (0.10) | −1.98 (0.06) | 239.35 | −0.63 (0.08) | −0.21 (0.09) | 240.34 | −0.54 (0.06) | −2.22 (0.06) |
| F$^*$ | 243.72 | −0.06 (0.10) | −2.04 (0.06) | 241.82 | −0.30 (0.08) | 0.04 (0.09) | 241.79 | −0.25 (0.06) | −2.15 (0.06) |
| F$^c$ | 285.17 | −0.45 (0.02) | −0.08 (0.03) | 286.73 | −0.74 (0.02) | −0.62 (0.02) | 291.75 | −0.75 (0.05) | −0.48 (0.04) |
| F$^{c*}$ | 293.74 | −0.01 (0.02) | −0.09 (0.03) | 291.17 | −0.02 (0.02) | 0.02 (0.02) | 294.46 | −0.01 (0.04) | −0.03 (0.03) |
| Faerosol | −5.74 | −0.21 (0.01) | 0.06 (0) | −2.47 | −0.33 (0.01) | −0.26 (0) | −1.49 | −0.29 (0.01) | −0.07 (0.01) |
| Faerosol$^c$ | −8.58 | −0.45 (0) | 0.01 (0) | −4.44 | −0.73 (0) | −0.65 (0) | −2.70 | −0.74 (0.01) | −0.45 (0.01) |
| Fcloud | −47.19 | 0.19 (0.11) | −1.90 (0.05) | −47.38 | 0.11 (0.07) | 0.41 (0.08) | −51.42 | 0.21 (0.05) | −1.74 (0.05) |
| Fcloud$^*$ | −50.02 | −0.05 (0.11) | −1.95 (0.06) | −49.35 | −0.28 (0.05) | 0.02 (0.09) | −52.67 | −0.24 (0.05) | −2.12 (0.05) |
| dFcloud | 2.83 | 0.24 (0.01) | 0.06 (0) | 1.97 | 0.39 (0) | 0.39 (0) | 1.25 | 0.44 (0.01) | 0.38 (0.01) |
| AOD | 0.144 | 0.032 (0) | 0 (0) | 0.103 | 0.025 (0) | 0.021 (0) | 0.102 | 0.027 (0.001) | 0.018 (0.01) |
| COD | 9.545 | 0.051 (0.019) | 1.985 (0.016) | 8.881 | 0.164 (0.021) | −0.016 (0.022) | 9.666 | 0.092 (0.012) | 0.580 (0.023) |
| CLD | 54.22 | 0.03 (0.07) | −0.07 (0.03) | 62.42 | 0.04 (0.05) | −0.01 (0.07) | 63.83 | 0.02 (0.05) | 0.29 (0.05) |
| LWP | 65.32 | −0.24 (0.15) | 11.56 (0.09) | 75.11 | 0.29 (0.21) | 0.31 (0.23) | 40.93 | −0.20 (0.10) | 3.53 (0.10) |
| CDNC | 1.65 | −0.01 (0) | 0.84 (0) | 1.82 | 0 (0) | 0 (0) | 0.99 | 0 (0) | 0.39 (0) |

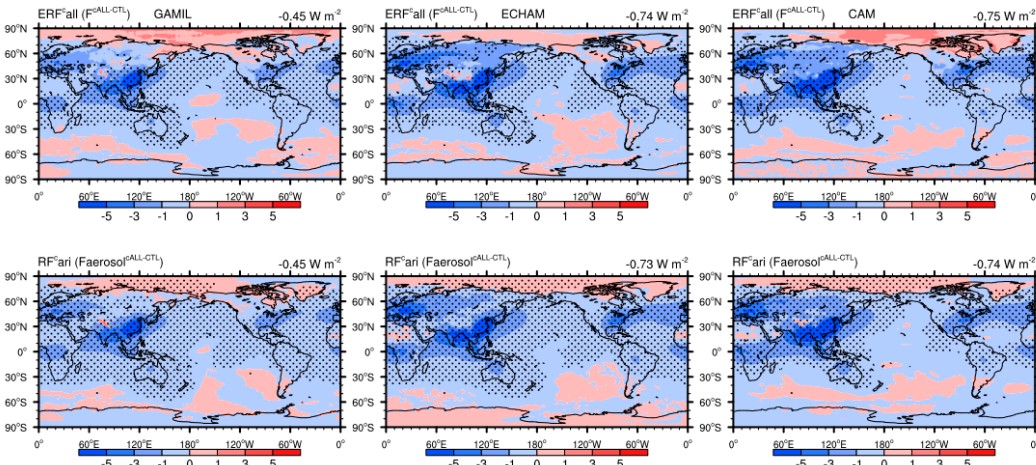

**Figure 3.** Same as Figure 2, but for clear-sky effective radiative forcing (ERF$^c$all = F$^{cALL-CTL}$, upper panel) and clear-sky instantaneous radiative forcing from aerosol–radiation interactions (RF$^c$ari = Faerosol$^{cALL-CTL}$, lower panel).

Here, the anthropogenic aerosol climate effects (combined "ari" and "aci") based on the CMIP6 protocol are compared with those based on the models' own default treatments. Table 3 lists the differences between the CTL and ALL experiments and the differences between the CTL and OLD experiments. Firstly, we analyzed the GAMIL model experiments. The AOD$^{ALL-CTL}$ and AOD$^{OLD-CTL}$ were 0.032 and 0, respectively. The default GAMIL model does not consider anthropogenic aerosol direct radiative forcing. Thus, the Faerosol$^{ALL-CTL}$ was −0.21 W·m$^{-2}$, whereas the Faerosol$^{OLD-CTL}$ was 0.06 W·m$^{-2}$. Note that the positive value of Faerosol$^{OLD-CTL}$ was caused by the impact of anthropogenic aerosol "aci" on diagnosing Faerosol. This is well discussed in Section 3.4. Because only the Twomey effect is considered in the MACv2-SP, the relative changes in the column-integrated grid-mean cloud droplet number concentration (CDNC$^{ALL-CTL}$) and the liquid water path (LWP$^{ALL-CTL}$) were very small. However, CDNC$^{OLD-CTL}$ and LWP$^{OLD-CTL}$ were obvious because a stronger Twomey effect and subsequent lifetime effect are considered in the default GAMIL model. This is one reason for the fact that the COD$^{ALL-CTL}$ (0.051) was clearly less than the COD$^{OLD-CTL}$ (1.985). As expected, the Fcloud$^{*ALL-CTL}$ (−0.05 W·m$^{-2}$) was clearly weaker than the Fcloud$^{*OLD-CTL}$ (−1.95 W·m$^{-2}$). Hence, the ERFall estimated based on the CMIP6 protocol (F$^{ALL-CTL}$, −0.27 W·m$^{-2}$) was clearly weaker than that based on the model's own default treatment (F$^{OLD-CTL}$, −1.98 W·m$^{-2}$). Secondly, we analyzed the ECHAM model experiments. The AOD$^{ALL-CTL}$ (0.025) was a little larger than the AOD$^{OLD-CTL}$ (0.021). This explains why the Faerosol$^{ALL-CTL}$ (−0.33 W·m$^{-2}$) and Faerosol$^{cALL-CTL}$ (−0.73 W·m$^{-2}$) were stronger than the Faerosol$^{OLD-CTL}$ (−0.26 W·m$^{-2}$) and Faerosol$^{cOLD-CTL}$ (−0.65 W·m$^{-2}$). Because the default ECHAM model does not consider the aerosol indirect effect, the Fcloud$^{*OLD-CTL}$ was almost zero (0.02 W·m$^{-2}$). However, the Fcloud$^{*ALL-CTL}$ was −0.28 W·m$^{-2}$. Thus, the ERFall estimated based on the CMIP6 protocol (F$^{ALL-CTL}$, −0.63 W·m$^{-2}$) was clearly stronger than that based on the model's own default approach (F$^{OLD-CTL}$, −0.21 W·m$^{-2}$). Finally, we analyzed the CAM model experiments. The AOD$^{ALL-CTL}$ (0.027) was larger than the AOD$^{OLD-CTL}$ (0.018). This is the primary reason for the fact that the Faerosol$^{ALL-CTL}$ (−0.29 W·m$^{-2}$) was stronger than the Faerosol$^{OLD-CTL}$ (−0.07 W·m$^{-2}$). However, the Fcloud$^{*ALL-CTL}$ (−0.24 W·m$^{-2}$) was clearly weaker than the Fcloud$^{*OLD-CTL}$ (−2.12 W·m$^{-2}$). As a result, the ERFall estimated from the MACv2-SP (F$^{ALL-CTL}$, −0.54 W·m$^{-2}$) was clearly weaker than that from the model's own default approach (F$^{OLD-CTL}$, −2.22 W·m$^{-2}$). In short, the difference in ERFall based on the models' default approaches among these three models (−1.98 W·m$^{-2}$ GAMIL, −0.21 W·m$^{-2}$ ECHAM, −2.22 W·m$^{-2}$ CAM) was clearly greater than that based on the CMIP6 protocol. The model diversity in ERFall was dramatically reduced after using the same anthropogenic aerosol forcing. The ERFall based on the CMIP6 protocol from the ECHAM (−0.63 W·m$^{-2}$) and CAM (−0.54 W·m$^{-2}$) models fell within the range of ERFall with the five

climate models (from −0.9 to −0.4 W·m$^{-2}$) shown in Fiedler et al. [29]. However, the GAMIL model produced a very weak ERFall (−0.27 W·m$^{-2}$). Excluding this outlier, the difference in ERFall between the ECHAM and CAM models was 0.10 W·m$^{-2}$, which was much less than that based on the models' own default approaches (2.01 W·m$^{-2}$).

### 3.3. Contributions from "Ari"

As discussed in Section 2.4, there are two ways to estimate the contributions from "ari", i.e., the difference between the ARI and CTL experiments and the difference between the ALL and ACI experiments. Table 4 lists these two kinds of changes. For all models, under clear-sky conditions, the global annual mean F$^{cARI-CTL}$ was almost identical to the F$^{cALL-ACI}$. In other words, the values for ERF$^c$ari calculated by the above two methods were almost identical. Furthermore, the standard deviations of ERF$^c$ari (i.e., F$^{cARI-CTL}$ and F$^{cALL-ACI}$) from all model experiments were very small. After considering cloudy skies, the standard deviations of ERFari (i.e., F$^{ARI-CTL}$ and F$^{ALL-ACI}$) were clearly increased. Therefore, the F$^{ARI-CTL}$ might be quite different from F$^{ALL-ACI}$, e.g., the F$^{ARI-CTL}$ (−0.21 W·m$^{-2}$) from the GAMIL model was 50% stronger than the F$^{ALL-ACI}$ (−0.14 W·m$^{-2}$). This indicates that the radiative effect of cloud-relevant rapid adjustment is the main contributor to the perturbation of modeled ERFari. The averaged ERFari results (0.5F$^{ARI-CTL}$ + 0.5F$^{ALL-ACI}$) from the GAMIL, ECHAM, and CAM models were −0.18, −0.28, and −0.23 W·m$^{-2}$, respectively. The maximum difference among these three models was 0.10 W·m$^{-2}$.

**Table 4.** Same as Table 3, but shown are changes from aerosol-radiation interactions (i.e., the ARI experiment minus the CTL experiment and the ALL experiment minus the ACI experiment).

| | GAMIL | | ECHAM | | CAM | |
|---|---|---|---|---|---|---|
| | **ARI −CTL** | **ALL −ACI** | **ARI −CTL** | **ALL −ACI** | **ARI −CTL** | **ALL −ACI** |
| F | −0.21 (0.05) | −0.14 (0.05) | −0.25 (0.07) | −0.31 (0.10) | −0.24 (0.08) | −0.21 (0.08) |
| F$^*$ | 0 (0.05) | 0.07 (0.06) | 0.09 (0.08) | 0.03 (0.11) | 0.07 (0.09) | 0.08 (0.08) |
| F$^c$ | −0.45 (0.02) | −0.45 (0.02) | −0.71 (0.02) | −0.72 (0.02) | −0.74 (0.04) | −0.73 (0.03) |
| F$^{c*}$ | 0 (0.02) | 0 (0.02) | 0.02 (0.02) | 0.01 (0.02) | 0 (0.04) | 0.01 (0.04) |
| Faerosol | −0.21 (0) | −0.21 (0) | −0.35 (0) | −0.34 (0.01) | −0.31 (0.01) | −0.30 (0.01) |
| Faerosol$^c$ | −0.45 (0) | −0.45 (0) | −0.73 (0) | −0.73 (0) | −0.74 (0.01) | −0.74 (0.01) |
| Fcloud | 0.24 (0.05) | 0.30 (0.05) | 0.46 (0.07) | 0.41 (0.10) | 0.50 (0.07) | 0.51 (0.08) |
| Fcloud$^*$ | 0 (0.05) | 0.07 (0.05) | 0.08 (0.07) | 0.02 (0.10) | 0.06 (0.07) | 0.08 (0.08) |
| dFcloud | 0.24 (0) | 0.24 (0) | 0.38 (0) | 0.39 (0) | 0.43 (0) | 0.44 (0.01) |
| AOD | 0.032 (0) | 0.032 (0) | 0.025 (0) | 0.025 (0) | 0.027 (0) | 0.027 (0) |
| COD | −0.037 (0.022) | −0.044 (0.012) | −0.034 (0.017) | −0.011 (0.020) | −0.018 (0.020) | −0.027 (0.026) |
| CLD | 0.07 (0.06) | 0.01 (0.05) | −0.03 (0.07) | 0.02 (0.10) | −0.01 (0.04) | 0 (0.06) |
| LWP | −0.20 (0.10) | −0.23 (0.09) | 0.17 (0.17) | 0.36 (0.19) | −0.04 (0.08) | −0.11 (0.12) |
| CDNC | 0 (0) | −0.01 (0) | 0 (0) | 0 (0) | 0 (0) | 0 (0) |

The Faerosol$^{ALL-ACI}$ and Faerosol$^{cALL-ACI}$ were almost identical to the Faerosol$^{ARI-CTL}$ and Faerosol$^{cARI-CTL}$ (not shown). This again indicates that the diagnosed RFari and RF$^c$ari can be well constrained. Here, only the RFari (Faerosol$^{ARI-CTL}$) and RF$^c$ari (Faerosol$^{cARI-CTL}$) from the ARI and CTL experiments are shown in Figure 4. It is obvious that both RFari and RF$^c$ari from the GAMIL model ($-0.21$ and $-0.45$ W·m$^{-2}$) were substantially weaker than those from the ECHAM ($-0.35$ and $-0.73$ W·m$^{-2}$) and CAM ($-0.31$ and $-0.74$ W·m$^{-2}$) models. The main reason for this is that the all-sky and clear-sky natural aerosol radiative forcings (Faerosol and Faerosol$^c$) from the GAMIL model ($-5.74$ and $-8.58$ W·m$^{-2}$) were stronger (more negative) than those from the ECHAM ($-2.47$ and $-4.44$ W·m$^{-2}$) and CAM ($-1.49$ and $-2.70$ W·m$^{-2}$) models (Table 3). A stronger Faerosol (Faerosol$^c$) can result in a weaker modeled RFari (RF$^c$ari) [33]. A sensitivity test (not introduced in this study) showed that the RFari from the GAMIL model can be enhanced to $-0.30$ W·m$^{-2}$ by reducing the background natural aerosol radiative effect.

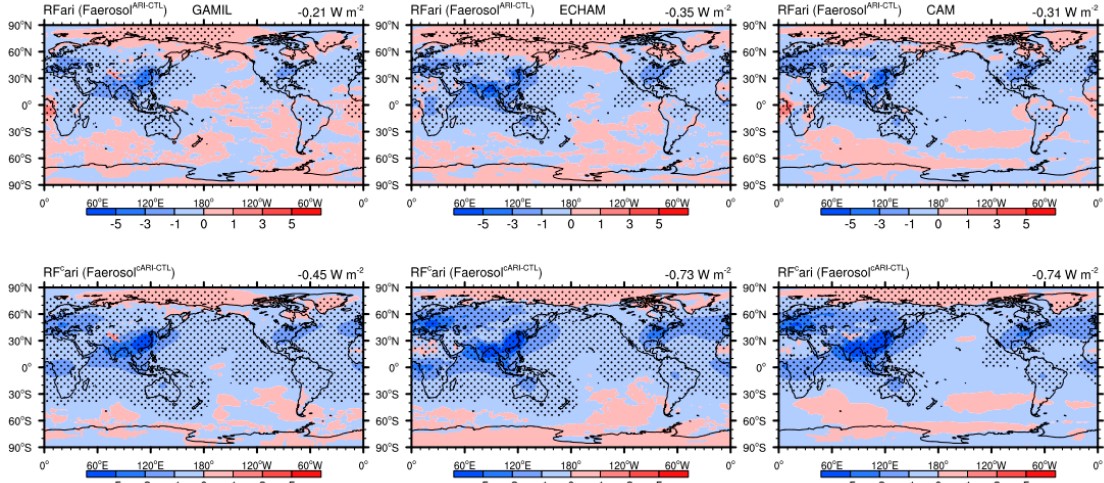

**Figure 4.** Anthropogenic aerosol all-sky (RFari = Faerosol$^{ARI-CTL}$) and clear-sky (RF$^c$ari = Faerosol$^{cARI-CTL}$) instantaneous radiative forcing from the differences between the ARI and CTL experiments with the GAMIL (left), ECHAM (middle), and CAM (right) models. The number on the top right of the figure denotes the ensemble average of global annual mean values. Differences significant at the 10% level of the Student's *t*-test are depicted by dots.

The semi-direct effects from all models are shown in Figure 5. Both Fcloud$^{*ARI-CTL}$ and Fcloud$^{*ALL-ACI}$ can represent the semi-direct effect. The global mean semi-direct effect ranged from 0 to 0.08 W·m$^{-2}$. The positive sign is consistent with the fact that the rapid adjustments reduce the ERF of black carbon [54]. As compared with RFari, the spatial distribution of the semi-direct effect was very disorderly. The regional distribution of Fcloud$^{*ARI-CTL}$ was not close to Fcloud$^{*ALL-ACI}$. There were few regions that could pass the 10% significance level test. In terms of global mean values, the magnitudes of Fcloud$^{*ARI-CTL}$ and Fcloud$^{*ALL-ACI}$ were statistically non-significant as compared with their standard deviations (Table 4). Taking the ECHAM model experiments, for example, the global means for Fcloud$^{*ARI-CTL}$ and Fcloud$^{*ALL-ACI}$ were 0.08 and 0.02 W·m$^{-2}$, respectively. These values were not clearly larger than the corresponding standard deviations (0.07 and 0.10 W·m$^{-2}$). The differences between Fcloud$^{*ARI-CTL}$ and Fcloud$^{*ALL-ACI}$ from the GAMIL (0.07 W·m$^{-2}$) and ECHAM (0.06 W·m$^{-2}$) models were very noticeable. This suggests that 10 member ensembles with the 10-year averages are not enough to get a stable estimate of semi-direct effect for these two models.

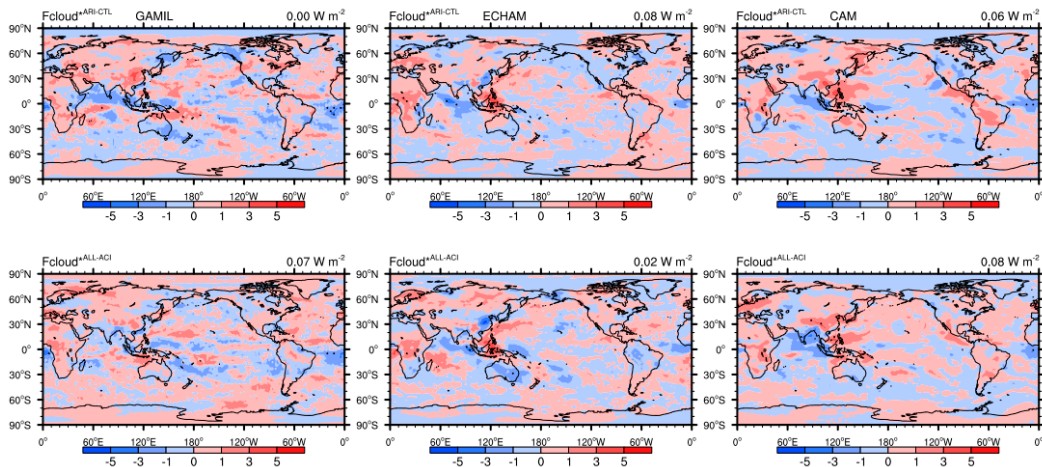

**Figure 5.** Anthropogenic aerosol effects on shortwave cloud forcing from aerosol–radiation interactions with the GAMIL (left), ECHAM (middle), and CAM (right) models. The upper panel (Fcloud$^{*ARI-CTL}$) shows the differences in shortwave cloud forcing between the ARI and CTL experiments. The lower panel (Fcloud$^{*ALL-ACI}$) shows the differences between the ALL and ACI experiments. The number on the top right of the figure denotes the ensemble average of global annual mean values. Differences significant at the 10% level of the Student's *t*-test are depicted by dots.

### 3.4. Contributions from "Aci"

Like "ari", there are also two ways to estimate the contributions from "aci" (Table 5). For all models, the changes in Fcloud were very close to the changes in Fcloud$^*$, because of the fact that "ari" was excluded. Note that the Faerosol might be non-zero, e.g., the Faerosol$^{ALL-ARI}$ from the ECHAM model was 0.02 W·m$^{-2}$ and larger than its standard deviation (0.01 W·m$^{-2}$). The reason for this is that "aci" could also impact the diagnosis of Faerosol. Compared with the impact of "ari" on diagnosing Fcloud, this impact was very small and negligible. It is obvious that ERFaci mainly depended on the changes in Fcloud or Fcloud$^*$. The standard deviations of the changes in Fcloud, Fcloud$^*$, and F were noticeable. As a result, the F$^{ACI-CTL}$ may be clearly different from F$^{ALL-ARI}$. The averaged ERFaci (0.5F$^{ACI-CTL}$ + 0.5F$^{ALL-ARI}$) from the GAMIL, ECHAM, and CAM models were −0.09, −0.35, and −0.32 W·m$^{-2}$, respectively. The maximum difference in the averaged ERFaci among the three models was 0.26 W·m$^{-2}$. This was the dominant source of the model diversity in ERFall as compared with ERFari.

**Table 5.** Same as Table 3, but shown are changes from aerosol-cloud interactions (i.e., the ACI experiment minus the CTL experiment and the ALL experiment minus the ARI experiment).

|  | GAMIL | | ECHAM | | CAM | |
|---|---|---|---|---|---|---|
|  | **ACI −CTL** | **ALL −ARI** | **ACI −CTL** | **ALL −ARI** | **ACI −CTL** | **ALL −ARI** |
| F | −0.12 (0.08) | −0.06 (0.13) | −0.32 (0.08) | −0.37 (0.06) | −0.33 (0.07) | −0.30 (0.05) |
| F$^*$ | −0.12 (0.08) | −0.06 (0.14) | −0.32 (0.08) | −0.39 (0.06) | −0.33 (0.07) | −0.32 (0.06) |
| F$^c$ | −0.01 (0.02) | −0.01 (0.02) | −0.02 (0.02) | −0.03 (0.02) | −0.02 (0.05) | −0.01 (0.04) |
| F$^{c*}$ | −0.01 (0.02) | 0 (0.02) | −0.02 (0.02) | −0.03 (0.02) | −0.02 (0.04) | −0.02 (0.04) |

**Table 5.** *Cont.*

| | GAMIL | | ECHAM | | CAM | |
|---|---|---|---|---|---|---|
| | ACI −CTL | ALL −ARI | ACI −CTL | ALL −ARI | ACI −CTL | ALL −ARI |
| Faerosol | 0 (0) | 0 (0.01) | 0.01 (0) | 0.02 (0.01) | 0 (0.01) | 0.02 (0.01) |
| Faerosol$^c$ | 0 (0) | 0 (0) | 0 (0) | 0 (0) | 0 (0.01) | 0.01 (0.01) |
| Fcloud | −0.12 (0.07) | −0.06 (0.14) | −0.29 (0.08) | −0.34 (0.06) | −0.31 (0.05) | −0.29 (0.06) |
| Fcloud$^*$ | −0.12 (0.08) | −0.06 (0.15) | −0.30 (0.08) | −0.36 (0.06) | −0.31 (0.05) | −0.30 (0.06) |
| dFcloud | 0 (0) | 0 (0.01) | 0.01 (0) | 0.02 (0) | 0.01 (0.01) | 0.01 (0) |
| AOD | 0 (0) | 0 (0) | 0 (0) | 0 (0) | 0 (0.001) | 0 (0.001) |
| COD | 0.096 (0.017) | 0.089 (0.034) | 0.175 (0.017) | 0.198 (0.016) | 0.120 (0.029) | 0.110 (0.012) |
| CLD | 0.02 (0.04) | −0.04 (0.10) | 0.02 (0.09) | 0.07 (0.09) | 0.02 (0.06) | 0.03 (0.04) |
| LWP | −0.01 (0.12) | −0.05 (0.22) | −0.08 (0.20) | 0.13 (0.16) | 0.03 (0.12) | −0.04 (0.06) |
| CDNC | 0 (0) | 0 (0.01) | 0 (0) | 0 (0) | 0 (0) | 0 (0) |

The spatial patterns of ERFaci were almost identical to changes in Fcloud$^*$ (not shown) because the ERFaci mainly depends on the Twomey effect. The anthropogenic aerosol effects on cloud forcing from the Twomey effect can be estimated as Fcloud$^{*ACI-CTL}$ or Fcloud$^{*ALL-ARI}$ (Figure 6). For all models, there were few regions that could pass the 10% significance level test. The main reason for these uncertainties is discussed in the next paragraph. For the ECHAM and CAM models, the spatial patterns of Fcloud$^{*ACI-CTL}$ were generally similar to Fcloud$^{*ALL-ARI}$, but one should keep in mind that the local Fcloud$^{*ACI-CTL}$ and Fcloud$^{*ALL-ARI}$ were almost not statistically significant. The common negative regions were located over high anthropogenic aerosol burden areas (dN > 1.25, Figure 1). As compared with the ECHAM and CAM models, the spatial patterns of Fcloud$^{*ACI-CTL}$ and Fcloud$^{*ALL-ARI}$ from the GAMIL model seem disordered. This phenomenon might be explained by the weak Twomey effect. In terms of global mean values, the Fcloud$^{*ACI-CTL}$ and Fcloud$^{*ALL-ARI}$ from the GAMIL model (−0.12 and −0.06 W·m$^{-2}$) were clearly weaker than that from the ECHAM (−0.30 and −0.36 W·m$^{-2}$) and CAM (−0.31 and −0.30 W·m$^{-2}$) models. The Twomey effect works by changing cloud optical depth (COD). The COD$^{ACI-CTL}$ and COD$^{ALL-ARI}$ from the ECHAM model (0.175 and 0.198) were clearly larger than that from the GAMIL (0.096 and 0.089) and CAM (0.120 and 0.110 W·m$^{-2}$) models. This is not consistent with the difference in the Twomey effect among these three models. Furthermore, both the GAMIL and CAM models use a two-moment stratiform cloud microphysics scheme, whereas a one-moment cloud microphysics scheme is used in the ECHAM model. This is also not consistent with the difference in the Twomey effect among these three models. It seems that the reason for the model diversity in the Twomey effect is very complex and difficult to figure out.

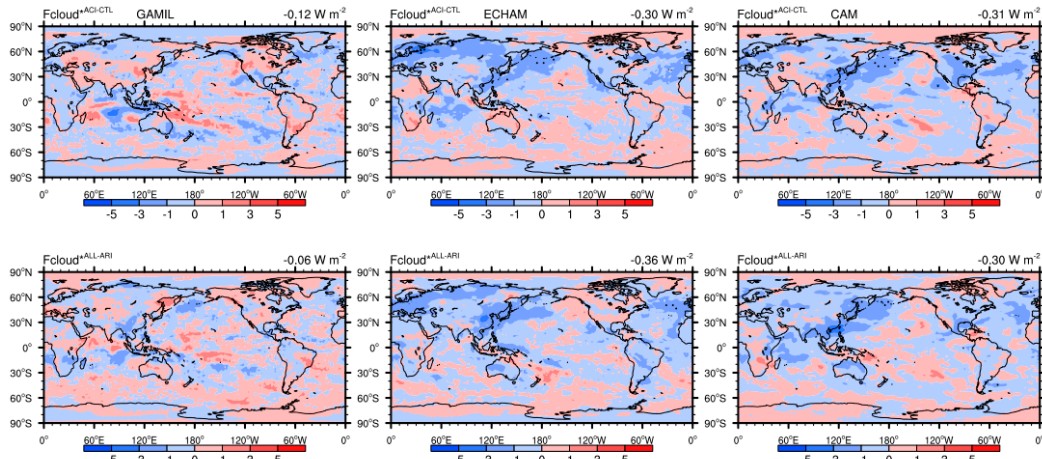

**Figure 6.** Anthropogenic aerosol effects on shortwave cloud forcing from Twomey effect with the GAMIL (left), ECHAM (middle), and CAM (right) models. The upper panel (Fcloud$^{*\text{ACI}-\text{CTL}}$) shows the differences in shortwave cloud forcing between the ACI and CTL experiments. The lower panel (Fcloud$^{*\text{ALL}-\text{ARI}}$) shows the differences between the ALL and ARI experiments. The number on the top right of the figure denotes the ensemble average of global annual mean values. Differences significant at the 10% level of the Student's *t*-test are depicted by dots.

The standard deviations of Fcloud$^{*\text{ACI}-\text{CTL}}$ and Fcloud$^{*\text{ALL}-\text{ARI}}$ from the GAMIL model were 0.08 W·m$^{-2}$ and 0.15 W·m$^{-2}$, respectively. These standard deviations were close to their corresponding ensemble averages (−0.12 and −0.06 W·m$^{-2}$). The Fcloud$^{*\text{ACI}-\text{CTL}}$ and Fcloud$^{*\text{ALL}-\text{ARI}}$ not only include the instantaneous Twomey effect (i.e., RFaci), but also subsequent changes in cloud optical properties from rapid adjustments. It is necessary to figure out the relative contribution of RFaci and rapid adjustments. Figure 7 shows RFari from the ACI and ALL experiments. Note that both RFaci$^{\text{CTL}}$ and RFaci$^{\text{ARI}}$ were zero. The RFaci$^{\text{ACI}}$ and RFaci$^{\text{ALL}}$ showed identical magnitudes and regional distributions. The global magnitude of RFaci was −0.10 W·m$^{-2}$. There were only negative regions of RFaci, and the maximum negative region was located in east and south Asia and adjacent oceans. All regions of RFaci were statistically significant. In short, the modeled RFaci was very stable. This indicates that the rapid adjustments induced by the Twomey effect contributed to the standard deviations of ERFaci, Fcloud$^{*\text{ACI}-\text{CTL}}$, and Fcloud$^{*\text{ALL}-\text{ARI}}$. In terms of global mean, the sign of this rapid adjustment was not clear (Fcloud$^{*\text{ACI}-\text{CTL}}$ − RFaci$^{\text{ACI}}$ = −0.02 W·m$^{-2}$ or Fcloud$^{*\text{ALL}-\text{ARI}}$ − RFaci$^{\text{ALL}}$ = 0.04 W·m$^{-2}$). It is interesting to point out that the impact of model internal variability on RFari was obvious (Figures 2 and 4) as compared with RFaci. The reason is that RFari is derived from the difference between two experiments (e.g., Faerosol$^{\text{ARI}-\text{CTL}}$), whereas RFaci is derived from one experiment (e.g., RFaci$^{\text{ACI}}$). As compared with RFaci, RFari includes the impact of the differences in background atmospheric state between two experiments caused by model internal variability.

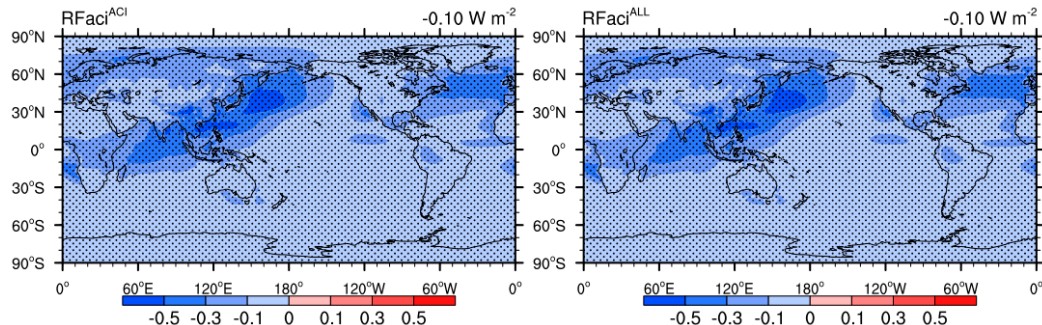

**Figure 7.** Instantaneous aerosol Twomey effect (RFaci) from the ACI (left) and ALL (right) experiments with the GAMIL model. The number on the top right of the figure denotes the ensemble average of global mean values. Differences significant at the 10% level of the Student's *t*-test are depicted by dots.

## 4. Conclusions

The same anthropogenic aerosol forcings dataset, which provides aerosol optical properties (i.e., aerosol–radiation interactions, "ari") and the associated change in cloud droplet numbers (i.e., aerosol–cloud interactions, "aci"), were implemented into three climate models, i.e., the GAMIL, ECHAM, and CAM models. One goal of this study was to test whether the diversity in modeled anthropogenic aerosol effective radiative forcing (ERF) was reduced. The ensemble average of anthropogenic aerosol (year 2000) ERF (ERFall) was estimated at $-0.27$ W·m$^{-2}$ with the GAMIL model, $-0.63$ W·m$^{-2}$ with the ECHAM model, and $-0.54$ W·m$^{-2}$ with the CAM model (Table 6). The difference in ERFall among these three models was clearly reduced as compared with the ERFall based on the models' own default approaches ($-1.98$ W·m$^{-2}$ GAMIL, $-0.21$ W·m$^{-2}$ ECHAM, $-2.22$ W·m$^{-2}$ CAM, Table 6).

**Table 6.** Ensemble averages of effective radiative forcing (ERF) and its two basic components, i.e., the instantaneous radiative forcing from aerosol–radiation interactions (RFari) and the aerosol-induced changes in cloud forcing. Standard deviations (in brackets) are calculated from the different ensemble members.

| Names (Calculating Methods) | GAMIL | ECHAM | CAM |
|---|---|---|---|
| ERFall ($F^{OLD-CTL}$) | $-1.98$ (0.06) | $-0.21$ (0.09) | $-2.22$ (0.06) |
| ERFari (Faerosol$^{OLD-CTL}$) | 0.06 (0) | $-0.26$ (0) | $-0.07$ (0.01) |
| Fcloud$^{*OLD-CTL}$ | $-1.95$ (0.06) | 0.02 (0.09) | $-2.12$ (0.05) |
| ERFall ($F^{ALL-CTL}$) | $-0.27$ (0.10) | $-0.63$ (0.08) | $-0.54$ (0.06) |
| RFari (Faerosol$^{ALL-CTL}$) | $-0.21$ (0.01) | $-0.33$ (0.01) | $-0.29$ (0.01) |
| Fcloud$^{*ALL-CTL}$ | $-0.05$ (0.11) | $-0.28$ (0.05) | $-0.24$ (0.05) |
| ERFari ($F^{ARI-CTL}$) | $-0.21$ (0.05) | $-0.25$ (0.07) | $-0.24$ (0.08) |
| ERFari ($F^{ALL-ACI}$) | $-0.14$ (0.05) | $-0.31$ (0.10) | $-0.21$ (0.08) |
| ERFari ($0.5F^{ARI-CTL} + 0.5F^{ALL-ACI}$) | $-0.18$ | $-0.28$ | $-0.23$ |
| RFari (Faerosol$^{ARI-CTL}$) | $-0.21$ (0) | $-0.35$ (0) | $-0.31$ (0.01) |
| RFari (Faerosol$^{ALL-ACI}$) | $-0.21$ (0) | $-0.34$ (0.01) | $-0.30$ (0.01) |
| RFari ($0.5$Faerosol$^{ARI-CTL} +$ $0.5$Faerosol$^{ALL-ACI}$) | $-0.21$ | $-0.35$ | $-0.31$ |
| Fcloud$^{*ARI-CTL}$ | 0 (0.05) | 0.08 (0.07) | 0.06 (0.07) |
| Fcloud$^{*ALL-ACI}$ | 0.07 (0.05) | 0.02 (0.10) | 0.08 (0.08) |
| $0.5$Fcloud$^{*ARI-CTL} + 0.5$Fcloud$^{*ALL-ACI}$ | 0.04 | 0.05 | 0.07 |
| ERFari ($F^{ACI-CTL}$) | $-0.12$ (0.08) | $-0.32$ (0.08) | $-0.33$ (0.07) |
| ERFari ($F^{ALL-ARI}$) | $-0.06$ (0.13) | $-0.37$ (0.06) | $-0.30$ (0.05) |
| ERFari ($0.5F^{ACI-CTL} + 0.5F^{ALL-ARI}$) | $-0.09$ | $-0.35$ | $-0.32$ |
| Fcloud$^{*ACI-CTL}$ | $-0.12$ (0.08) | $-0.30$ (0.08) | $-0.31$ (0.05) |
| Fcloud$^{*ALL-ARI}$ | $-0.06$ (0.15) | $-0.36$ (0.06) | $-0.30$ (0.06) |
| $0.5$Fcloud$^{*ACI-CTL} + 0.5$Fcloud$^{*ALL-ARI}$ | $-0.09$ | $-0.33$ | $-0.31$ |

After using the same prescribed anthropogenic aerosol forcings dataset, the still existing differences among these three participating models were also analyzed. The modeled ERFall can be decomposed into two main contributors, i.e., instantaneous radiative forcing from "ari" (RFari) and aerosol-induced changes in cloud forcing (Fcloud$^{*ALL-CTL}$). The RFari from the GAMIL, ECHAM, and CAM models were $-0.21$ W·m$^{-2}$, $-0.33$ W·m$^{-2}$, and $-0.29$ W·m$^{-2}$, respectively (Table 6). The main reason for the model diversity in RFari was the differences in natural aerosol radiative forcings ($-5.74$ W·m$^{-2}$ GAMIL, $-2.47$ W·m$^{-2}$ ECHAM, $-1.49$ W·m$^{-2}$ CAM, Table 3). The RFari from the GAMIL model could be enhanced to $-0.30$ W·m$^{-2}$ by decreasing the background natural aerosol (not shown). In other words, the model diversity in RFari could be constrained by reducing the differences in natural aerosol. The Fcloud$^{*ALL-CTL}$ from the GAMIL, ECHAM, and CAM models were $-0.05$ W·m$^{-2}$, $-0.28$ W·m$^{-2}$, and $-0.24$ W·m$^{-2}$, respectively (Table 6). Compared to RFari, the difference in Fcloud$^{*ALL-CTL}$ was the dominant source of the model diversity in ERFall. However, it was difficult to figure out the reason for the model diversity in Fcloud$^{*ALL-CTL}$ as compared with RFari.

In order to deeply understand the sources for model diversity in Fcloud$^{*ALL-CTL}$, the two components of Fcloud$^{*ALL-CTL}$ (i.e., semi-direct effect and the Twomey effect) were analyzed. The semi-direct effect can be estimated as Fcloud$^{*ARI-CTL}$ or Fcloud$^{*ALL-ACI}$. The global mean semi-direct effects ($0.5$Fcloud$^{*ARI-CTL}$ + $0.5$Fcloud$^{*ALL-ACI}$) were estimated at $0.04$ W·m$^{-2}$ with GAMIL, $0.05$ W·m$^{-2}$ with ECHAM, and $0.07$ W·m$^{-2}$ with CAM. These global mean values were not clearly larger than their standard deviations (Table 6). In short, the semi-direct effect was small and variable, and the model diversity in the global mean semi-direct effect was rather small. The Twomey effect can be estimated as Fcloud$^{*ACI-CTL}$ or Fcloud$^{*ALL-ARI}$. The global mean Twomey effects ($0.5$Fcloud$^{*ACI-CTL}$ + $0.5$Fcloud$^{*ALL-ARI}$) were estimated at $-0.09$ W·m$^{-2}$ with GAMIL, $-0.33$ W·m$^{-2}$ with ECHAM, and $-0.31$ W·m$^{-2}$ with CAM (Table 6). It is clear that the model diversity in Fcloud$^{*ALL-CTL}$ was predominantly caused by the Twomey effect. Unfortunately, there are a lot of atmospheric tunable parameters that can cause uncertainty in the simulated change in cloud radiative state, and it is not easy to constrain these tunable parameters, to a large extent [8]. Therefore, the reason for the model diversity in Fcloud$^{*ALL-CTL}$ might be very complex.

In this study, ensemble simulations were carried out to estimate a more stable ERF and to analyze the impact of model internal variability. The GAMIL model experiments show that there is almost no impact of model internal variability on RFaci (Figure 7). This suggests that the RF estimated from double radiation calls is not sensitive to model internal variability. A single simulation can produce a robust RF. The model internal variability influences the ERF through rapid adjustments. The rapid adjustments can be grouped according to different sources as either the semi-direct effect or rapid adjustment of the Twomey effect. In this study, the global mean semi-direct effects (from 0 to $0.08$ W·m$^{-2}$) were not clearly larger than their standard deviations (Table 6). One model experiment shows that the sign of global mean rapid adjustment caused by the Twomey effect was not clear ($-0.02$ or $0.04$ W·m$^{-2}$, Section 3.4). In short, for each individual model, although the contribution from rapid adjustments to the means of modeled ERF is small, the rapid adjustments have an important role to play in the quantification of the perturbation of ERF.

Understanding the difference between the aerosol climate effects we want and the aerosol climate effects we calculate is very important. Taking anthropogenic aerosol instantaneous radiative forcing from "ari" (RFari) for example, the RFari should be diagnosed from the difference between two radiation calls with and without anthropogenic aerosol optical properties at each radiation time step. In other words, the RFaci should be derived from one experiment. In this study, the RFari was approximated by the difference in Faerosol between two simulations with and without anthropogenic aerosols (i.e., Faerosol$^{ALL-CTL}$, Faerosol$^{ARI-CTL}$, or Faerosol$^{ALL-ACI}$). This will lead to a notable impact of model internal variability on RFari. Another more noteworthy example is the Twomey effect. The normal diagnosed Twomey effect (i.e., difference in Fcloud$^{*}$ between two simulations with and without the Twomey effect, Fcloud$^{*ACI-CTL}$ or Fcloud$^{*ALL-ARI}$) not only includes the instantaneous Twomey effect (i.e., the definition of Twomey effect, RFaci), but also subsequent changes in cloud optical properties

from rapid adjustments. Therefore, in this study, calculation methods for estimating aerosol climate effects were introduced in detail.

**Author Contributions:** X.S. designed the study and wrote large portions of the manuscript. W.Z. and J.L. contributed to some writing of the manuscript. All model experiments were performed and analyzed by X.S. All authors contributed to the discussion of model results.

**Funding:** This study was jointly supported by the National Key Research and Development Program of China (grant nos. 2017YFA0604001) and the National Natural Science Foundation of China (grant nos. 41775095).

**Acknowledgments:** The authors would like to thank Bjorn Stevens for the guidance in implementing the MACv2-SP parameterization into climate models.

**Conflicts of Interest:** The authors declare no conflict of interest.

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
