# Peer review of "Comparison of Anthropogenic Aerosol Climate Effects among Three Climate Models with Reduced Complexity"

_atmosphere, doi:10.3390/atmos10080456_

Round 1

Reviewer 1 Report

I write in regards to Manuscript entitled "Comparison of Anthropogenic Aerosol Climate Effects among Three Climate Models with Reduced Complexity" which you submitted to Atmosphere.

I have read this paper that deals with the comparison of the anthropogenic aerosol climate effects among three different climate models, the GAMIL, ECHAM and CAM models. In general, the paper is concise, clear and well presented. In my opinion this work is well organized and the results shown are valid, and I suggest accepting this manuscript as it is in Atmosphere.

Author Response

We thank the reviewer for the positive comments. The reviewer’s comments are in italics and our responses in the standard fond below.

General Assessment:

I have read this paper that deals with the comparison of the anthropogenic aerosol climate effects among three different climate models, the GAMIL, ECHAM and CAM models. In general, the paper is concise, clear and well presented. In my opinion this work is well organized and the results shown are valid, and I suggest accepting this manuscript as it is in Atmosphere.

Reply: Thanks very much for the positive comments.

Comments:

None

Please see the attachment (revised manuscript).

Reviewer 2 Report

This article implements the simplified MACv2-SP aerosol parameterization into three climate models to test whether a single, simple aerosol scheme applied across models reduces the spread in estimates of effective radiative forcing. The findings largely support the hypothesis that ERF spread is reduced compared to experiments with the model's own aerosol treatments. The analysis is fairly detailed, but is limited mainly to global averages and spatial maps of differences between pairs of simulations. The strength of the analysis is that instantaneous radiative forcing and "clean sky" fluxes are calculated with double radiation calls. The experimental design is also good, in particular the series of ARI, ACI, and ALL is very nice for separating the direct and Twomey effects. There are some minor weaknesses in the paper, as detailed below, which should be addressed. The overarching theme of these comments is that I think the text could be simplified and made more clear and the main results could be more succinct.

Comments

1. First, I think the text is somewhat misleading about the role of MACv2-SP in CMIP6. It is not the recommendation for modeling centers to use MACv2-SP as the main way aerosol effects are treated in the models. This is implied, however, in several places (Lines 11-12, 43-46, 480-481). The correct statement, as far as I am aware, is given at lines 58-59, stating that CMIP6 encourages participation in RFMIP, and RFMIP includes using MACv2-SP in some experiments.

2. In the experimental design, it is unclear why pairs of experiments with the same initialization date used for differences. This is described in the paragraph at lines 169-180. There are 10 ensemble members for each experiment, and then 10 differences are made between each set of experiments based on initialization date. Although the results later in the paper seem fairly robust, this seems like an opportunity to have a much more robust estimate of ERF by doing all 100 comparisons between simulations.

3. Perhaps the weakest point of the paper is the notation. I was frequently confused and flipping back to definitions to try to keep straight all the different quantities. Table 1 is invaluable, and Table 2 is very helpful. I think at the very least another table is needed that provides the definitions of the RFx and ERFx. I'm also still very unclear about why Fari (line 199) is ever used, as it seems like it is synonymous with ERFari and Faci and ERFaci seem to be the same. Maybe Fari and Faci mean just 'ari' and 'aci', but it does not quite seem like that. Similarly, the Delta notation in 217-228 and then 229-246 is somewhat clumsy. The meaning is clear, but it seems like there must be a more obvious way to do this. In fact, I am not sure why the Delta is introduced, since the previous paragraph had introduced the notation of using the superscript to explicitly define the difference. That notation takes a lot of space, but at least it is clear. I do not understand why \Delta SWCF* is preferred to SWCF*^{ALL-ACI}. I suppose it is because the Delta can stand for two different comparisons (ALL-ACI and ARI-CTL). The fact is that the notation is difficult manage, and there is not an obvious answer. Including another table, more clearly defining Fari and Faci, and possibly getting rid of the Deltas could all help to streamline the reading experience. 

4. One additional note on notation. The use of GAMIL/CAM variable names for the fluxes (e.g., FSNT) might be hard for some readers who are not used to them. The positive attribute that they have is that the are just capital characters, but the negative is that they are not obvious ("flux shortwave net top-of-model"). It might be easier to use the CMIP versions of the variable names (so more readers would recognize them), or to use some other symbol (like just F for net shortwave flux; \tau for optical depth; N for CDNC). This is more minor than #3, but if the notation is being reconsidered, it might be worth keeping in mind.

5. I think the conclusions section could use a revision to be more concise and clear. It has a lot of good information, but it seems a little too long and it does not clearly state the main conclusions. One possible addition that might help is to include one more table that has ERFall, RFari, SWCF*ALL-CTL (and optionally "Twomey") as columns with the three models as the rows. This would get the main quantitative results into one small space. That might provide an easier way to describe the results. Right now these numbers are inline throughout the page starting at Line 483. 

Technical

Line 12: implicated -> implemented

Line 330: lift-time -> lifetime

Line 408-409: "also impact the diagnosis of SWAF"

Author Response

We thank the reviewer for the constructive comments and suggestions for improving this manuscript. We hope that the modified manuscript and our response to the comments are satisfactory. The reviewer’s comments are in italics and our responses in the standard fond below.

General Assessment:

This article implements the simplified MACv2-SP aerosol parameterization into three climate models to test whether a single, simple aerosol scheme applied across models reduces the spread in estimates of effective radiative forcing. The findings largely support the hypothesis that ERF spread is reduced compared to experiments with the model's own aerosol treatments. The analysis is fairly detailed, but is limited mainly to global averages and spatial maps of differences between pairs of simulations. The strength of the analysis is that instantaneous radiative forcing and "clean sky" fluxes are calculated with double radiation calls. The experimental design is also good, in particular the series of ARI, ACI, and ALL is very nice for separating the direct and Twomey effects. There are some minor weaknesses in the paper, as detailed below, which should be addressed. The overarching theme of these comments is that I think the text could be simplified and made more clear and the main results could be more succinct.

Reply: Thanks very much for the comments. In the revised version, to simplify the text, figure 7 (zonal mean ERFaci and COD changes) and its corresponding text were removed. In the conclusions section, the motivations of this study are clearly pointed out and the main conclusions are concise. We have addressed all these weaknesses with point-by-point responses as detailed below.

Comments:

1. First, I think the text is somewhat misleading about the role of MACv2-SP in CMIP6. It is not the recommendation for modeling centers to use MACv2-SP as the main way aerosol effects are treated in the models. This is implied, however, in several places (Lines 11-12, 43-46, 480-481). The correct statement, as far as I am aware, is given at lines 58-59, stating that CMIP6 encourages participation in RFMIP, and RFMIP includes using MACv2-SP in some experiments.

Reply: Following the reviewer’s comment, "CMIP6" was removed in the abstract and conclusions sections (i.e., lines 11-12, 480-481 in previous version).

         The paper "Overview of the Coupled Model Intercomparison Project Phase 6 (CMIP6) experimental design and organization" introduced the CMIP6 forcing data sets as follows:

"The historical forcings are based as far as possible on observations and cover the period 1850–2014. These include:

– for simulations with prescribed aerosols, a new approach to prescribe aerosols in terms of optical properties and fractional change in cloud droplet effective radius to provide a more consistent representation of aerosol forcing, and

Some models might require additional forcing data sets (e.g. black carbon on snow or anthropogenic dust). Allowing model groups to use different forcing data sets might better sample uncertainty, but makes it more difficult to assess the uncertainty in the response of models to the best estimate of the forcing, available to a particular CMIP phase. To avoid conflating uncertainty in the response of models to a given forcing, it is strongly preferred for models to be integrated with the same forcing in the entry card historical simulations".

The CMIP6 website (https://www.wcrp-climate.org/wgcm-cmip/wgcm-cmip6) also introduced the CMIP6 forcing. "The CMIP6 datasets needed for the DECK, historical, and CMIP6-Endorsed MIP experiments are briefly described at http://goo.gl/r8up31". "Aerosol Optical Properties and Relative Change in Cloud Droplet Number Concentration – Contact: Bjorn Stevens ([email protected]). To provide a more consistent representation of aerosol forcing for simulations with prescribed aerosols, the officially recommended CMIP6 aerosol forcing dataset is the simple plume aerosol climatology described below. This is a new approach where aerosols are prescribed in terms of optical properties and relative change in cloud droplet number concentration".

I am not sure whether it is the recommendation for modeling centers to use the same prescribed anthropogenic aerosol forcing data (i.e., MACv2-SP) according to the CMIP6 experimental design. It is quite certain that both the GAMIL and ECHAMNUIST models with MACv2-SP are used in CMIP6.

2. In the experimental design, it is unclear why pairs of experiments with the same initialization date used for differences. This is described in the paragraph at lines 169-180. There are 10 ensemble members for each experiment, and then 10 differences are made between each set of experiments based on initialization date. Although the results later in the paper seem fairly robust, this seems like an opportunity to have a much more robust estimate of ERF by doing all 100 comparisons between simulations.

Reply: In the same experiment, the differences in 10-year average radiative fluxes between two ensemble members (i.e., two simulations with the different initialization date) are notable due to model-internal variability. It is better to exclude this kind of differences from estimating ERF. This is the reason why pairs of experiments with the same initialization date used for differences. We added this note in the revised manuscript.

     Ensemble simulations can produce the standard deviation of ERF. The standard deviation is used to indicate the possible range of ERF. In other words, a smaller standard deviation indicates a more robust estimate of ERF. The standard deviation calculated from all 100 differences between any simulations might be larger than that calculated from the 10 differences between simulations with the same initialization date. Thus, 100 comparisons between simulations cannot produce a more robust estimate of ERF.

3. Perhaps the weakest point of the paper is the notation. I was frequently confused and flipping back to definitions to try to keep straight all the different quantities. Table 1 is invaluable, and Table 2 is very helpful. I think at the very least another table is needed that provides the definitions of the RFx and ERFx. I'm also still very unclear about why Fari (line 199) is ever used, as it seems like it is synonymous with ERFari and Faci and ERFaci seem to be the same. Maybe Fari and Faci mean just 'ari' and 'aci', but it does not quite seem like that. Similarly, the Delta notation in 217-228 and then 229-246 is somewhat clumsy. The meaning is clear, but it seems like there must be a more obvious way to do this. In fact, I am not sure why the Delta is introduced, since the previous paragraph had introduced the notation of using the superscript to explicitly define the difference. That notation takes a lot of space, but at least it is clear. I do not understand why \Delta SWCF* is preferred to SWCF*^{ALL-ACI}. I suppose it is because the Delta can stand for two different comparisons (ALL-ACI and ARI-CTL). The fact is that the notation is difficult manage, and there is not an obvious answer. Including another table, more clearly defining Fari and Faci, and possibly getting rid of the Deltas could all help to streamline the reading experience.

Reply: Sorry for the confusion. In the revised manuscript, 'Fari' and 'Faci' were replaced by 'ari' and 'aci'. Following the reviewer’s comment, the Delta was removed. The "SWCF" was replaced by "Fcloud". Both Fcloud*ARI-CTL and Fcloud*ALL-ACI were shown in this study (Table 4 and Figure 5). The definitions of the RFx and ERFx are pretty clear and easy to remember. However, the methods for calculating the RFx and ERFx might not be unique. Another table that provides all calculation methods for estimating the RFx and ERFx was added (Table 6). Furthermore, in all figures, the methods for calculating the RFx and ERFx were shown on the top left of the figure, such as ERFall (FALL−CTL). In the text, the methods for calculating the RFx and ERFx were usually shown in subsequent brackets, such as ERFari (i.e., FARI−CTL and FALL−ACI).

4. One additional note on notation. The use of GAMIL/CAM variable names for the fluxes (e.g., FSNT) might be hard for some readers who are not used to them. The positive attribute that they have is that the are just capital characters, but the negative is that they are not obvious ("flux shortwave net top-of-model"). It might be easier to use the CMIP versions of the variable names (so more readers would recognize them), or to use some other symbol (like just F for net shortwave flux; \tau for optical depth; N for CDNC). This is more minor than #3, but if the notation is being reconsidered, it might be worth keeping in mind.

Reply: Thanks for the reviewer’s comment. In the revised manuscript, we use the capital letter "F" to indicate the all-sky shortwave net radiative fluxes at TOA. The F marked with the superscript "*" (i.e., F*) is diagnosed from radiation call with aerosol scattering and absorption neglected. The F marked with the superscript "c" (i.e., Fc) is the clear-sky F, which is diagnosed from radiation call without cloud effect. The shortwave aerosol forcing and shortwave cloud forcing are named as the Faerosol and Fcloud, respectively. Faerosol=F-F* and Fcloud=F-Fc. We also added this note in the revised manuscript.

5. I think the conclusions section could use a revision to be more concise and clear. It has a lot of good information, but it seems a little too long and it does not clearly state the main conclusions. One possible addition that might help is to include one more table that has ERFall, RFari, SWCF*ALL-CTL (and optionally "Twomey") as columns with the three models as the rows. This would get the main quantitative results into one small space. That might provide an easier way to describe the results. Right now these numbers are inline throughout the page starting at Line 483.

Reply: Thanks for the reviewer’s comment. We rewrote the conclusions section. In the revised manuscript, the motivations of this study are clearly pointed out and the main conclusions are concise. Furthermore, another table that shows the ERF andits two basic components (i.e., the RFari and the difference in cloud forcing) was added (Table 6).

Technical:

Line 12: implicated -> implemented

Reply: Done.

Line 330: lift-time -> lifetime

Reply: Done.

Line 408-409: "also impact the diagnosis of SWAF"

Reply: Done.

We also upload the revised manuscript (see the attachment).

Reviewer 3 Report

Manuscript ID: atmosphere-548759

Title: Comparison of Anthropogenic Aerosol Climate Effects among Three > Climate Models with Reduced Complexity

Authors: Xiangjun Shi, Wentao Zhang, Jiaojiao Liu

The manuscript is focussed on simulations of anthropogenic aerosol effects on the cloud droplet number concentration (Twomey’s effect) carried out by three climate models with reduced complexity. The manuscript is designed to fill the gap for understanding model differences in estimating anthropogenic aerosol effects, searching for uncertainty sources that are not clearly defined in more complex models.

The manuscript is well written, and I enjoyed reading it - although in some places it seems to lose legibility (for example, in the paragraph in which the method for estimating ERF from Fari is introduced - page 6, lines 217-228). Similarly, for estimating ERF from Faci - page 6, lines 229-246.

However, a description of the statistical analysis methods used to compare the results of the models is missing.

The description of the results appears to be rather complex, but an adequate discussion of the results is lacking. As it is, the reader perceives an arid description of the results and, when there is, a comparison of results not supported by explicit and rigorous statistical tests.

I, therefore, suggest:

To reduce the description of the models in the materials and methods paragraph, adding the details considered of interest to the reader in the appendix;

To describe the statistical methods used to compare the results modelled through the three models and to define the range of estimation errors for each model;

To insert in the results also a discussion (not to be left in the conclusions) that also takes into account the results obtained by other authors, taken from the current bibliography.

Author Response

We would like to thank the reviewer for the constructive comments. We hope that the modified manuscript gives a more clear explanation of our results. The reviewer’s comments are in italics and our responses in standard fond below.

General Assessment:

The manuscript is focussed on simulations of anthropogenic aerosol effects on the cloud droplet number concentration (Twomey’s effect) carried out by three climate models with reduced complexity. The manuscript is designed to fill the gap for understanding model differences in estimating anthropogenic aerosol effects, searching for uncertainty sources that are not clearly defined in more complex models.

The manuscript is well written, and I enjoyed reading it - although in some places it seems to lose legibility (for example, in the paragraph in which the method for estimating ERF from Fari is introduced - page 6, lines 217-228). Similarly, for estimating ERF from Faci - page 6, lines 229-246.

However, a description of the statistical analysis methods used to compare the results of the models is missing.

The description of the results appears to be rather complex, but an adequate discussion of the results is lacking. As it is, the reader perceives an arid description of the results and, when there is, a comparison of results not supported by explicit and rigorous statistical tests.

Reply: Thanks very much for the comments. In the revised manuscript, we rewrote the paragraphs in which the methods for estimating ERFari and ERFaci are introduced. The description of the results is simplified. For example, figure 7 (zonal mean ERFaci and COD changes) and its corresponding text were removed. We added some discussions about why the reason for the model diversity in the Twomey effect is very complex and difficult to figure out. We clearly pointed out the necessity of understanding the difference between the aerosol climate effects we want and the aerosol climate effects we calculate.  

Comments:

1. To reduce the description of the models in the materials and methods paragraph, adding the details considered of interest to the reader in the appendix.

Reply: This study analyzes the differences in estimating aerosol climate effects among these three participating models with reduced complexity. In order to deeply understand the sources for model diversity, some readers might want to know about the detail of model descriptions and calculation methods. In the revised manuscript, we also clearly introduced the importance of model descriptions and calculation methods, such as "Note that the radiation package of GAMIL (Delta–Eddington Approximation) is different to that of ECHAM and CAM (RRTMG). As a result, the visible band of GAMIL (350.0–640.0 nm) is different to that of ECHAM and CAM (441.5–625.0 nm). This explains why AODa from GAMIL is slightly higher than that from ECHAM and CAM". 

2. To describe the statistical methods used to compare the results modelled through the three models and to define the range of estimation errors for each model.

Reply: In recent studies about estimating aerosol climate effects with climate model, ensemble simulations are usually carried out to obtain a reliable estimate of ERF. Because the number of ensemble members is small (usually less than 12), the standard deviation calculated from the different ensemble members is used to indicate the possible range of ERF (e.g. Stevens et al., 2017; Fiedler et al., 2019). Furthermore, as far as I know, the results from different models are compared directly if the number of participating models is small (e.g. Fiedler et al., 2019, five models). The analytical methods used in the study are similar to those used in Fiedler et al. (2019).

References

Stevens, B.; Fiedler, S.; Kinne, S.; Peters, K.; Rast, S.; Müsse, J.; Smith, S.J.; Mauritsen, T. MACv2-SP: A parameterization of anthropogenic aerosol optical properties and an associated Twomey effect for use in CMIP6. Geosci. Model Dev. 2017, 10, 433–452.

Fiedler, S.; Kinne, S.; Huang, W.T.K.; Räisänen, P.; amp; apos; Donnell, D.; Bellouin, N.; Stier, P.; Merikanto, J.; et al. Anthropogenic aerosol forcing—Insights from multiple estimates from aerosol-climate models with reduced complexity. Atmos. Chem. Phys. 2019, 19, 6821–684

3. To insert in the results also a discussion (not to be left in the conclusions) that also takes into account the results obtained by other authors, taken from the current bibliography.

Reply: Following the reviewer’s comment, in the revised manuscript, we added some discussions in the results section, such as " The global mean semi-direct effect ranges from 0 to 0.08 W m–2. The positive sign is consistent with the fact that the rapid adjustments reduce the ERF of black carbon [54]". In the results section, we compared our modelled ERF with the results from Fiedler et al. (2019). I am quite certain that this study is the first to show the modelled ERF using the GAMIL model with reduced complexity (i.e., MACv2-SP). As far as I am aware, there is no published study that shows the modelled ERF using the CAM model with MACv2-SP. In shorts, there are few studies that show modelled ERF from multi-models with MACv2-SP, except for the study of Fiedler et al. (2019).

We also upload the revised manuscript (see the attachment).

Round 2

Reviewer 3 Report

The manuscript is much improved compared to the previous version as a result of the authors following the suggestions made by the reviewer.

In the Materials and Methods section, the authors did not specify the statistical methods used for comparing the outputs of the models, as required by the previous reviewer. TThe word "significant" is not supported by any statistical test. Therefore, the authors are asked to comply with this request.

Author Response

We would like to thank the reviewer for the comments. We hope that the modified manuscript gives a more clear explanation of our results. The reviewer’s comments are in italics and our responses in standard fond below.

General Assessment:

The manuscript is much improved compared to the previous version as a result of the authors following the suggestions made by the reviewer.

Reply: Thanks very much for the comments.

Comments:

1. In the Materials and Methods section, the authors did not specify the statistical methods used for comparing the outputs of the models, as required by the previous reviewer. TThe word "significant" is not supported by any statistical test. Therefore, the authors are asked to comply with this request.

Reply: In our study, all modelled aerosol climate effects (i.e., RFs, ERFs, and changes in cloud forcing) are statistically tested, following Fiedler et al. 2017. Significance tests are performed with Student’s t-tests. We clearly pointed out this statistical method in the revised manuscript (changes are marked in blue). Furthermore, same "significant" are replaced by "obvious", such as "The global mean ERFalls from the GAMIL, ECHAM, and CAM models are −0.27, −0.63, and −0.54 W m–2, respectively. The ERFall from the GAMIL model is obviously weaker (less negative) than that from the ECHAM and CAM models".